# The Effects of Light and the Circadian System on Rhythmic Brain Function

**DOI:** 10.3390/ijms23052778

**Published:** 2022-03-03

**Authors:** Charlotte von Gall

**Affiliations:** Institute of Anatomy II, Medical Faculty, Heinrich Heine University, 40225 Dusseldorf, Germany; charlotte.vongall@med.uni-duesseldorf.de

**Keywords:** suprachiasmatic nucleus, entrainment, masking, chronodisruption, molecular clockwork, behavior, cognition, hippocampus, light at night, phase shift, melatonin, glucocorticoids, circadian rhythms, circadian clock, clock genes

## Abstract

Life on earth has evolved under the influence of regularly recurring changes in the environment, such as the 24 h light/dark cycle. Consequently, organisms have developed endogenous clocks, generating 24 h (circadian) rhythms that serve to anticipate these rhythmic changes. In addition to these circadian rhythms, which persist in constant conditions and can be entrained to environmental rhythms, light drives rhythmic behavior and brain function, especially in nocturnal laboratory rodents. In recent decades, research has made great advances in the elucidation of the molecular circadian clockwork and circadian light perception. This review summarizes the role of light and the circadian clock in rhythmic brain function, with a focus on the complex interaction between the different components of the mammalian circadian system. Furthermore, chronodisruption as a consequence of light at night, genetic manipulation, and neurodegenerative diseases is briefly discussed.

## 1. Introduction

Life on earth has evolved under the influence of rhythmic changes in the environment, such as the 24 h light/dark cycle. Living organisms have developed internal circadian clocks, which allow them to anticipate these rhythmic changes and adapt their behavior and physiology accordingly. This is most obvious for plants in which the anticipation of the time window for photosynthesis, the light phase, provides a selection advantage over plants that simply react to the onset of the light phase. Moreover, for the early cold-blooded terrestrial animals, anticipating the light phase and, thus, the time with higher ambient temperature and better availability of visual cues has been a selection advantage. In contrast, for the early (warm-blooded) mammals, which developed in the Mesozoic (about 250 million years ago), anticipating the twilight and the dark phase, thus the time window for avoiding the diurnal predatory dinosaurs was crucial. Presumably, the early mammals gradually extended their behavior from the nocturnal towards the twilight phases of the day, resulting in activation of both cone- and rod-based vision [1]. Consequently, the neocortex, a brain region characteristic for mammals, which is responsible for higher-order brain functions, such as sensory perception, cognition, as well as the planning, control, and execution of voluntary movement, initially developed in nocturnal/crepuscular species. Only in the Cenozoic, when many species, including the non-avian dinosaurs, became extinct, mammals were released from this predatory pressure and diurnality developed among mammals [2]. Primates are among the earliest mammals to exhibit strict diurnal activity, approximately 52–33 million years ago [2]. Hence, mammals are by default nocturnal, and diurnalty as in humans is a relatively new invention and more or less an exception among the mammalian species. However, the visual system in primates is highly flexible and can function under bright and dim light conditions, hence allows evolutionary switching of lineages from one activity pattern to the other, according to the selective pressure [3]. About 40% of all mammal species are rodents. Among them there are very few diurnal species, such as Arvicanthis, Psammomys, and Ictidomys (formerly Spermophilus) [4]. Importantly, most rodents, including *Mus musculus* and *Rattus norwegicus*, the most commonly used mammals in the laboratory, are nocturnal. However, the aspect of different temporal niches is often not sufficiently taken into consideration when translating basic research in rodents into human applications. The aim of this broad review article is to highlight the role of light on rhythms in physiology and behavior, especially in nocturnal rodents from a neuroanatomical point of view, and to emphasize the important distinction between light-driven/time-of-day-dependent and endogenously driven/circadian rhythms.

## 2. The Role of Light and the Circadian Clock for Rhythmic Brain Function

### 2.1. The Mammalian Circadian System

In mammals, the circadian clock is hierarchically organized in a circadian system. The central circadian rhythm generator is located in the suprachiasmatic nucleus (SCN) of the hypothalamus. Rhythmic output of the SCN governs subsidiary circadian oscillators in the brain and the periphery. The SCN and the subsidiary oscillators consist of more or less strongly coupled cellular oscillators, each comprising a molecular clockwork composed of transcriptional/translational feedback loops of clock genes (reviewed in [5]). The SCN controls subsidiary circadian oscillators in the brain primarily via neuronal connections while peripheral oscillators are regulated via the rhythmic function of the autonomous nervous system [6] and the endocrine system [7]. The hormone of darkness, melatonin, and the stress hormone glucocorticoid [8] (see below) are important rhythmic signals for subsidiary circadian oscillators in the brain and in the periphery. The light input into the circadian system is provided by a subset of intrinsically photosensitive retinal ganglion cells (ipRGCs). There is increasing evidence that rhythmic light information is not only provided to the SCN but also directly or indirectly to many other brain regions, thus driving time-of day-dependent rhythmic brain function (Figure 1). 

### 2.2. Light Input into the Circadian System—Entrainment and Masking

Two main mechanisms help to specialize in a nocturnal or a diurnal niche [12]. In the mechanism called entrainment, light serves as a signal for the SCN, to match the period and the phase to the environmental oscillator, the light/dark regime. Adjusting the period is necessary, as the endogenous rhythm, persisting in constant darkness, is close to but not exact 24 h. The SCN in turn helps to anticipate the rhythmic changes in the environment and controls activity during the dark or the light phase. This is important, as many nocturnal mammals live in dark burrows and experience the environmental light conditions only when leaving the safe surrounding of the nest. Entrainment is adaptive, as the lengths of the light and dark phase change according to the seasons, depending on the latitude. In the mechanism called masking, light directly affects behavior obscuring the control from the circadian clock [13]. Masking is especially prominent in nocturnal species, and, here, light can have two opposite effects on activity depending on irradiance levels. In dim light, activity is increased compared to complete darkness. This enhancing effect of dim light, which is presumably due to increased confidence based on visual input [14], is called positive masking [12]. However, in complete darkness, activity is higher, despite the absence of visual cues, than under standard (bright) light conditions. This suppressive effect of bright light on activity is called negative masking [12]. Similarly, nocturnal animals prefer dark or dimly illuminated areas over brightly illuminated areas. This light aversion is strong enough to counteract the natural tendency to explore a novel environment, as shown by the light–dark test [15], a paradigm extensively used for tests on classic anxiolytics (benzodiazepines) [15], as well as anxiolytic-like compounds, such as serotonergic drugs or drugs acting on neuropeptide receptors (reviewed in [16]). In both diurnal and nocturnal mammals, light at night also elicits acute effects on physiological parameters, such as core body temperature and heart rate, as well as hormone secretion (see below). Importantly, the detection of visual information in the mammalian retina is conveyed by two parallel pathways, the rod–cone system for image-forming-vision and the melanopsin-based system of the ipRGCs for non-image-forming irradiance detection. Although ipRGCs are essential for the adaptive physiological responses to light, such as the pupillary light reflex [17], as well as circadian entrainment [18,19], and contribute to scotopic vision [20], both the rod–cone system and the ipRGCs seem to contribute to masking and light aversion [21]. Importantly, many commonly used laboratory mouse strains carry mutations that affect visual and/or non-visual physiology (reviewed in [22]).

Under natural conditions, entrainment and masking work in a complementary fashion [23]. However, in the laboratory, the two mechanisms can be segregated. As mentioned above, entrainment is highly adaptive to different photoperiods. In most animal facilities, the standard photoperiod is 12 h light and 12 h dark (LD 12:12), although some animal facilities have opted for different light conditions (e.g., LD 16:8), as the photoperiod has a high impact on reproduction in some species. The circadian clock also rapidly entrains to a phase shift one experiences when travelling across time zones or if the LD cycle is inverted. The re-entrainment capacity after jet lag is dependent on intrinsic factors, such as the robustness of the circadian clock or the signaling of hormones, such as melatonin [24]. Furthermore, the speed of re-entrainment to a phase shift depends on the direction; entrainment is usually faster in response to a phase delay than a phase advance [25]. Interestingly, this is the same in the diurnal human [26]. However, in nocturnal animals, brief light pulses during the early and late night are strong resetting cues for phase delays and advances of the circadian clock, respectively [27]. At the cellular level, photic resetting of the SCN molecular clockwork involves activation of the p44/42 mitogen-activated protein kinase (MAPK) signaling cascade, phosphorylation/activation of the transcription factor cAMP-response-element-binding protein (CREB) [28], the induction of the marker of neuronal activity c-Fos [29], inhibitors of DNA binding proteins [30], and expression of the clock gene *Per1* [31]. In order to study entrainment in the absence of the interfering masking effects of light, nocturnal animals can be housed under a so-called skeleton photoperiod, consisting of two discrete pulses of light during the early (dawn) and the late (dusk) light phase [32]. For entrainment to this lighting schedule, the intergeniculate leaflet is essential, which receives direct photic information from the ipRGCs [33].

### 2.3. The Brain Molecular Clockwork

Various brain functions, such as sleep, wake, foraging, food intake, alertness, emotion, motivation, and cognitive performance, controlled by different brain regions, show circadian rhythms. Moreover, any information is processed in a temporal context. Consistently, many brain regions harbour circadian oscillators, which are governed by the SCN [34]. At the cellular level, these oscillators are composed of single cells each harbouring a molecular clockwork composed of transcriptional/translational feedback loops of clock genes. The clock genes encode for activators of transcription, such as CLOCK and its forebrain-specific analog NPAS2 [35], BMAL1, and ROR, as well as the repressors of transcription PER1 and PER2, CRY1, CRY2, and REV-ERBα [5].

The molecular clockwork drives the rhythmic expression of clock controlled gene (see below) and posttranscriptional processes (reviewed in [36]) and modulates the chromatin landscape [37], thus regulating rhythmic cell function at multiple levels. The molecular clockwork in the SCN and subordinate extra-SCN brain circadian oscillators drives various rhythms in neuron and glia function including ATP concentration [38], neuronal electrical activity (reviewed in [39]), metabolism [40], redox homeostasis (reviewed in [41]), tyrosine hydroxylase expression in dopaminergic neurons [42], dopamine receptor signalling in the hippocampus [43], and extracellular glutamate homeostasis [44]. In addition, some rhythms in the SCN are time-of day-dependent and do not persist in constant darkness, such as rhythmic expression of connexion 30 [45], which contribute to astrocyte gap junctions and hemichannels (reviewed in [46]), as well as the stability of circadian rhythms and re-entrainment under challenging conditions [45]. Circadian clock gene expression in the SCN and the hippocampus persists with high robustness in vitro, indicating a strong coupling of single cell oscillators, while it damps rapidly in other brain regions, indicating a weak coupling [47,48,49]. Mice with a targeted deletion of the essential clock gene *Bmal1* are arrhythmic under constant environmental conditions [50], so a loss of function in a single gene strongly affects circadian rhythmicity. In mouse models for compromised molecular clockwork function, such as Bmal1-deficient mice, *Per1/2* double mutants, and *Cry1/Cry2* double mutants, circadian rhythms are abolished, while various parameters of physiology and behaviour are rhythmic under the LD 12:12 conditions due to masking [50,51,52]. This emphasizes the strong impact of the environmental light/dark conditions on rhythmic brain function. In this context, it is important to note that *Cry1/Cry2* double mutants and Bmal1-deficient mice show deficits in retinal visual physiology [53] and, consequently, impaired visual input into the circadian system [54,55]. Nevertheless, even under LD 12:12 conditions, many brain functions, such as spatial memory consolidation and contextual fear [56,57], adult neurogenesis [58], and sleep architecture [59] are affected in Bmal1-deficient mice, indicating the importance of this clock gene/transcription factor for general brain function. 

### 2.4. Rhythmic Gene and Protein Expression in the Brain

About 43% of all coding genes and about 1000 noncoding RNAs show circadian rhythms in transcription somewhere in the body, largely in an organ/tissue-specific manner [60,61,62]. The rhythmic transcriptome in peripheral organs is dependent on the SCN [62] but continues to oscillate in vitro for a few cycles [63]. Only 22% of circadian rhythmic mRNA is driven by de novo transcription, indicating that the molecular clock drives transcription and posttranslational modification [37]. Moreover, the epigenetic landscape is modulated in a circadian manner [37]. A comparable number of transcripts show a circadian oscillation in the SCN and the liver, while only about 10% of them show an overlap [61]. The core clock genes *Arntl* (encoding for Bmal1), *Dbp*, *Nr1d1* and *Nr1d2* (encoding for Rev-Erb alpha and beta, respectively), *Per1*, *Per2*, and *Per3,* as well as the clock controlled genes *Usp2*, *Tsc22d3*, and *Tspan4* oscillate in many organs and parts of the brain [60]. Importantly, many commonly used drugs target the products of the circadian genes, so the timed application of these drugs, chronotherapy, might maximize efficacy, and minimize side effects [60]. In accordance with the important role of the brain stem in the regulation of autonomous and vital functions, more than 30% of the drug-target circadian genes listed in the study by Zhang et al. (2014) are rhythmically expressed in this part of the brain. In the retina, about 277 genes show a circadian rhythm, implicated in a variety of functions, including synaptic transmission, photoreceptor signalling, intracellular communication, cytoskeleton reorganization, and chromatin remodelling [64]. Intriguingly, in LD 12:12, about 10 times as many genes oscillate, indicating that the LD cycle drives the rhythmic expression of a large number of genes in the retina [64]. In the forebrain synapses, a comparable amount of genes (2085, thus 67% of synaptic RNAs) show a time-of-day-dependent rhythm, and a high percentage of these genes remain rhythmic in constant darkness (circadian) [65]. Interestingly, the rhythmic genes in the forebrain synapses can be segregated into two temporal domains, predusk and predawn, relating to distinct functions; predusk mRNAs relate to synapse organization, synaptic transmission, cognition, and behaviour, while predawn mRNAs relate to metabolism, translation, and cell proliferation or development [65]. The oscillation of the synaptic proteome resembles those of the transcriptome [65] and a high percentage show an oscillation in the phosphorylation state [66].

#### Sleep Deprivation, Epilepsy, and Glucocorticoids Affect Gene and Protein Expression in the Brain

Sleep deprivation induced by gentle handling, cage tapping, and the introduction of novel objects during the light/inactive phase affects clock gene expression in the cerebral cortex [67] and leads to a reduction in transcript oscillation in the entire brain to about 20% [68]. This indicates that the sleep disruption itself, and/or the manipulation, as well as the associated additional light exposure, which mice usually do not experience while sleeping, strongly affects rhythmic transcription. On the other hand, it shows that only 20% of the rhythmic transcriptome in the brain is resilient to sleep deprivation, manipulation, and light exposure during the light/inactive phase. In forebrain synapses, sleep deprivation has a higher impact on the proteome and on rhythmic protein phosphorylation than on the transcriptome [65,66]. In this context, it is important to note that traditional sleep deprivation protocols using sensory-motor stimulation induces stress associated with a rise in circulating corticosterone [69], an important temporal signal within the circadian system (see below). Corticosterone strongly contributes to the sleep-deprivation-induced forebrain transcriptome [70]. Among the genes assigned to the corticosterone surge are clock genes, as well as genes implicated in sleep homeostasis, cell metabolism, and protein synthesis, while the transcripts that respond to sleep loss independent of corticosterone relate to neuroprotection [70]. The time-of-day-dependent oscillation in hippocampal transcriptome and proteome is affected by temporal lobe epilepsy [71]. Although epilepsy could be considered a chronic stress model [72], little is known on the contribution of glucocorticoids in these alterations. More research avoiding stress as a confounder is needed to explore the effect of sleep and neurological disorders on rhythmic brain function.

### 2.5. Circadian and Light-Driven Brain Function

#### 2.5.1. Rhythmic Hormone Release

The circadian rhythm of melatonin synthesis in the epithalamic pineal gland is one of the best characterized functions of the mammalian circadian system. The control of rhythmic melatonin synthesis comprises the rhythmic activation of the sympathetic nervous system by GABAergic neurons in the SCN projecting to pre-autonomous nerve cells in the paraventricular nucleus (PVN), and these neurons, in turn, project to preganglionic sympathetic neurons in the intermediolateral column of the spinal cord. The pineal gland is activated by postganglionic fibres from the superior cervical ganglia. Remarkably, in both nocturnal and diurnal species, the release of norepinephrine during the dark phase drives rhythmic melatonin synthesis and release. As rhythmic melatonin synthesis is governed by the circadian clock, it persists in constant darkness and can entrain to the environmental light/dark conditions. The duration of the melatonin signal increases with the length of the night, so melatonin provides a systemic signal not only for the phase of the night but also for anticipation and adaptation to seasonal changes in the photoperiod (reviewed in [10]), which is particularly relevant for seasonal breeders. Light, especially at a lower wavelength (<555 nm), during the dark phase acutely inhibits melatonin synthesis. The melatonin receptors MT1 and MT2, which belong to the superfamily of G-protein-coupled receptors, are widely distributed within the brain, including the SCN, and the periphery. Melatonin provides an important systemic time cue not only during adulthood but also during prenatal and early postnatal development when the components of the circadian system are not yet fully matured. During aging, the decrease in melatonin production and sensitivity is associated with an increasing deterioration of circadian rhythms. Interesting, many mouse strains, including those of the widely used C57BL/6 mice, do not produce melatonin as a result of spontaneous mutations [73], indicating that melatonin signalling is dispensable for living under laboratory conditions. In humans, melatonin has effects on sleep propensity, temperature regulation, and alertness and may modulate pain sensation, immune function, and metabolic function, such as insulin production (reviewed in [10]). Melatonin and melatonin receptor agonists are used for the treatment of jet lag symptoms, the entrainment of circadian rhythms in blind people, and major depression and insomnia, diseases considered to be associated with circadian dysfunction, as mentioned above. 

The circadian rhythm in glucocorticoid secretion from the adrenal provides an important systemic signal within the mammalian circadian system. Both the secretion in response to stress and the rhythmic basal secretion are regulated by the hypothalamo-pituitary-adrenal (HPA) axis. This comprises the release of corticotropine releasing hormone (CRH) from parvocellular neuroendocrine neurons in the PVN, which controls the secretion of adrenocorticotropic hormone (ACTH) from the anterior lobe of the pituitary into the systemic circulation and ACTH activates the release of glucocorticoids. The circadian rhythm of glucocorticoid is controlled by a vasopressinergic SCN projection to the PVN. In both diurnal and nocturnal animals, glucocorticoid levels start to rise in the second part of the inactive phase and reach peak values around wake time. Importantly, in nocturnal animals, brief light pulses during the subjective night leads to an increase in glucocorticoid levels comparable to those induced by a strong stressor [74]. Glucocorticoid receptors are widely distributed in the brain (except the SCN) and the body. Glucocorticoids are essential for life and regulate a variety of important cardiovascular, respiratory, metabolic, immunologic, and homeostatic functions. Interestingly, glucocorticoid signalling in utero is presumably a key mediator of prenatal stress and affects neurodevelopment and foetal epigenetic landscape [75]. Glucocorticoids have been shown to control various subsidiary clocks in the periphery, such as the liver, kidney, and heart [8] (reviewed in [76]). In addition, glucocorticoid signalling affects rhythmic gene expression in the brain regions implicated in emotions and cognition, such as the raphe nuclei, the amygdala, and the hippocampus [77,78,79,80]. 

#### 2.5.2. Rhythms in Food Intake

Although the regulation of food intake/foraging and energy metabolism strongly rely on homeostatic feedback signals, the circadian clock provides temporal organization (reviewed in [81]). It facilitates the temporal occurrence of related functions, such as food intake and glycogenesis, separates conflicting functions and behaviours, such as eating and sleep, and allows for the anticipation of rhythmic changes in the environment, such as the light/dark cycle of limited food availability (reviewed in [81]). An important homeostatic signal is the hormone ghrelin, released from gastric cells under fasting conditions, mediating the release of the neuropeptides neuropeptide Y and Agouti-related peptide from the arcuate nucleus of the hypothalamus. These neuropeptides, in turn, activate the release of orexin from the lateral hypothalamus (LH) and melanin-concentrating hormone. Orexin seems to play a major role in linking feeding behaviour and activity [82] (for the role of orexin in activity, see below). Anorexigenic humoral signals involve insulin and leptin, released from the pancreas and adipose tissue, respectively. Both signals converge on pro-opiomelanocortin expressing Acr neurons, and the α-melanocyte-stimulating hormone mediates hypophagic effects and increases in energy expenditure via the PVN, the dorsomedial hypothalamus (DMH), and the ventromedial hypothalamus (VMH) (reviewed in [81]). The homeostatic regulation of food intake and energy expenditure involves additional modulators, such as endocannabinoids and structures in the brain stem (reviewed in [81]). The SCN plays a major role for circadian rhythms in food intake [83]. In nocturnal animals, restricting the availability of food to the light phase affects many circadian rhythms: the animals show an increase in body temperature and glucocorticoid levels and become active a few hours in advance of the time of limited food availability. This rhythmic food anticipatory activity persists even under food deprivation for a couple of days, indicating an intrinsic time-keeping mechanism (reviewed in [84]). Curiously, this time-keeping mechanism of food anticipatory activity persists even if the SCN is disabled. However, the anatomical location of the so-called food entrainable oscillator is still unknown, and it might be a neuronal network rather than a single location (reviewed in [85]). Importantly, mistimed food intake has a variety of negative metabolic consequences, such as predisposition to obesity [86,87].

#### 2.5.3. The Sleep Wake Cycle

The sleep/wake cycle is the most prominent behavioural circadian rhythm. Sleep is also critically regulated by a homeostatic drive that increases with extended waking and dissipates by sleep (reviewed in [88]) and a complex process involving many brain regions and a network of wake- and sleep-promoting neurons (reviewed in [89], see below). During sleep, changes in cortical electrical activity, detectable by electroencephalography (EEG), occur and are classified as rapid eye movement (REM) sleep and non-REM sleep. The EEG during REM sleep is similar to the awake state, but with a loss of muscle tone, REMs, and active dreams. Non-REM sleep is divided into four stages representing a continuum of relative depth characterized by distinct EEG patterns and physiology. Stage 1 plays a role in the transition from wake to sleep, and stage 2 is characterized by mixed-frequency cortical activity and the presence of sleep spindles, which might be important for memory consolidation [90]. Stages 3 and 4 are collectively referred to as slow-wave sleep (SWS) because of the amplitude slow-wave cortical activity (reviewed in [91]). SWS plays an important role in the consolidation of hippocampus-dependent spatial memory [92]. During a sleep episode, REM sleep and non-REM sleep alternate in cycles (reviewed in [91]). Sleep has multiple functions besides memory consolidation and regeneration, including metabolite clearance (reviewed in [93]). In the current model, the major purpose of sleep is to restore structural and functional synapse homeostasis [94,95]. Moreover, the clearance of interstitial solutes in the brain, provided by the glial-lymphatic (=glymphatic) system, correlates with the prevalence of slow-wave sleep [96,97]. Importantly, glymphatic clearance might provide a link in the causal relationship between sleep disturbances and symptomatic progression in neurodegenerative diseases (reviewed in [98]). Sleep deprivation and insomnia have many negative consequences, including increased anxiety, decreased attention, and impaired executive function and cognitive performance [99,100]. Light has different effects on sleep and alertness in diurnal and nocturnal species (reviewed in [101]). In diurnal species, light increases arousal and alertness. In nocturnal species, the response also depends on the wavelength: blue light (470 nm) results in delayed sleep onset, light aversion, and elevated plasma corticosterone (see above), while green light (530 nm) of the same intensity leads to reduced arousal and sleep induction [102]. 

The brain circuitry that governs sleep and wakefulness/arousal include cell groups in the brain stem, hypothalamus, thalamus, and basal forebrain (reviewed in [103]) (Figure 2). The ascending reticular activating system (ARAS) in the brain stem is responsible for the control of wakefulness and sleep–wake transition. Cholinergic neurons of the ARAS activate the unspecific thalamus, which controls general cortical activity, and the specific thalamus, which controls the transmission of sensory information to the cortex. In addition, cortical activity is directly and indirectly modulated by a variety of brain stem nuclei, which employ different neurotransmitters, including the noradrenergic locus coeruleus, the dopaminergic ventral tegmental area, and the serotoninergic raphe nuclei. By interacting with other brain stem nuclei, the ARAS also modulates muscle tone, as well as autonomic functions, such as breathing, heart rate, and blood pressure during wake and sleep. Sound REM sleep is associated with a silencing of the locus coeruleus promoting synaptic plasticity (reviewed in [100]). Sleep- and wake-inducing hypothalamic nuclei control ARAS activity. During sleep, the ARAS is inhibited by a system of GABAergic neurons in which the ventrolateral preoptic nucleus (VLPO) of the hypothalamic preoptic region plays a key role [104]. Consistently, the largest class of sleep-promoting drugs/anaesthetics, including barbiturates, benzodiazepines, and chloral hydrate, enhances the activity of GABA receptors (reviewed in [103]). Orexin neurons in the lateral hypothalamus and histamine neurons in the tuberomamillary nucleus are mutually connected with the VLPO and the ARAS and synergistically regulate different aspects of the waking stage (reviewed in [105]). Orexin neurons project widely into other nuclei in the hypothalamus and into the forebrain, the thalamus, and the brain stem [106,107], indicating the complex role of the neuropeptide in autonomic, neuroendocrine, and cognitive function and emotion. Orexin might also convey an efferent signal to the food-entrainable oscillator [108]. An important driver of homeostatic sleep regulation is the neuromodulator adenosine, accumulating during wakefulness in the extracellular space as a by-product of neuronal metabolic activity [109]. Consistently, caffeine, the world‘s most widely consumed psychoactive drug, induces wakefulness, the release of norepinephrine, dopamine, and serotonin in the brain, and an increase in serum catecholamine levels by blocking adenosine receptors (reviewed in [110]). Glutamatergic neurons and, to a lesser extent, cholinergic neurons in the basal forebrain contribute to the adenosine-mediated control of sleep homeostasis [111]. For the control of the circadian rhythm in sleep/wakefulness projections of the SCN to the dorsomedial hypothalamus via the subparaventricular zone (SPZ) of the hypothalamus seem to play a major role (reviewed in [103]). Interestingly, the SPZ seems to have an amplifying and integrative role in the regulation of circadian rhythms in sleep, activity, and core body temperature, but with distinct subpopulations controlling the rhythms in body temperature or sleep/wake and locomotor activity [112]. Neurons in the dorsomedial hypothalamus project to the VLPO and the lateral hypothalamus using inhibitory and excitatory neurotransmitters orchestrating rhythmic changes in sleep–wake and wake–sleep transitions (reviewed in [103]). Importantly, the SPZ, the LH, and the VLPO receive direct innervation from the ipRGCs, so activation of the VLPO might account for light-induced sleep in nocturnal animals (reviewed in [113]). 

Sleep architecture is altered in mice with a compromised molecular clockwork even under LD 12:12 conditions. In Bmal1-deficient mice, the rhythms in total sleep time, REM and non-REM sleep, and core body temperature are blunted [59]. During sleep deprivation, Bmal1-defiecient mice show a reduced propensity for sustained wakefulness/higher sleep pressure and a reduced percentage of REM sleep during recovery [59]. Similarly, in *Cry1/Cry2* double mutants, the rhythms in REM and non-REM sleep are blunted and show high non-REM sleep pressure [67]. In *Per2* and *Per1/2* double mutants, the acrophase of rhythmic core body temperature is advanced, while the amplitude during the dark/active phase is reduced, and these mutants are more awake and have less REM sleep during the mid-third of the light phase [114]. Collectively, these data indicate an important role for clock genes in sleep pressure and sleep phase timing. Consistently, it has been suggested that insomnia is associated with polymorphisms in clock genes and clock-associated genes, such as peroxisome proliferator-activated receptor-γ coactivator-1α (PGC-1α), a rhythmically expressed transcriptional coactivator that regulates energy metabolism (reviewed in [100]).

#### 2.5.4. Cognitive Performance and Emotion-Related Behaviour

Cognitive performance, as well as brain functions affecting cognitive performance, such as mood/emotion, attention/arousal, sleep, core body temperature, and executive function, show time-of-day-dependence as well as circadian rhythms (reviewed in [101]). In humans, just after the nadir in body temperature shortly before wake time, sleepiness is highest, and vigilance and cognitive performance are lowest (reviewed in [115]), indicating a strong interconnection between these parameters. Studies using ‘forced desynchrony protocols’ in which subjects sleep in non-24 h schedules show circadian rhythms in cognitive performance, even when time spent awake has been controlled (reviewed in [116]), so sleep/wake and cognitive function are interconnected and independently regulated. However, a proper alignment between sleep/wake rhythms and internal circadian time is crucial for optimal cognitive performance [117]. 

Cognitive performance is a function of the neocortex and depends on sensory information processed by the specific thalamus. It is strongly regulated by paleocortical and archicortical input and by multiple projections from subcortical forebrain structures, the unspecific thalamus, and the brain stem, which convey emotional states, motivation, and alertness. Of note, the amygdala is a key structure in the forebrain for processing sensory information in the context of memory, decision making, and emotional responses, such as fear, anxiety, and aggression. It receives sensory information and input from other subcortical forebrain structures and the brain stem and sends projections to the entorhinal cortex (EC), modulating learning and memory (see below), to the hypothalamus, controlling acute, and chronic responses to stress, to the thalamus, controlling attention and alertness, and to the nucleus accumbens, controlling reward-related behaviour. 

Long-term memory is a three step process that consists of the acquisition of new information, consolidation of the acquired information, and the retrieval of stored information [118]. Importantly, in mice, long-term memory formation, especially training, is time-of-day-dependent with a peak during the early night [119]. Therefore, the time of day has a strong effect on the readout of tests on cognitive behaviour. The rhythm persists in constant darkness, indicating circadian-regulated memory consolidation [119]. In addition, light has a strong inhibitory effect on various cognitive functions and behavioural dimensions in mice [120,121]. Hence, for studies on cognitive function in nocturnal rodents, one should consider performing tests on behaviour and cognition in the dark phase (under LD 12:12) or in subjective night (under DD) and, thus, in the activity phase of the animals and without the disturbing influence of light. 

The archicortical hippocampus is the major brain region for episodic memory formation and for the integration of temporal and spatial information enabling navigation. It integrates sensory information, as well as information about the emotional and motivational state. Cholinergic, serotonergic, noradrenergic, and dopaminergic input for the brain stem modulates hippocampal function. The hippocampus includes the dentate gyrus (DG) and the cornu ammonis (CA), which is divided into four subfields (CA1–4). The dorsal hippocampus (DH) and the ventral hippocampus (VH) are functionally and anatomically distinct regions. The DH has a high density of so-called place cells, the cellular correlate for encoding spatial information (reviewed in [122]), and serves for spatial memory and conceptual learning, while the VH is strongly connected to the amygdala and implicated in stress responses, emotional behaviour, and contextual fear learning (reviewed in [123]). Processed sensory information reaches the CA1 pyramidal neurons via direct and indirect projections from the EC. The indirect projection, called the trisynaptic circuit, includes the projection from EC to DG granule cells and from there to CA3 pyramidal cells, sending their axons, called Schaffer collaterals, to the CA1. The hippocampus shows circadian rhythms in clock gene expression [124] that are almost 180 degrees out of phase with the expression rhythms measured from the SCN [48,80]. Mice with deletions/mutations of clock genes show impaired hippocampus-dependent memory formation [124] and a reduced ability to link spatial information with the time of day [125]. Moreover, hippocampus-dependent memory training leads to an upregulation of *Per1*, presumably via modulating the occupancy of the *Per1* promoter by the histone deacetylase HDAC3, which is also implicated in the age-related impairment of functional synaptic plasticity [126]. 

The cellular substrate for hippocampal learning and memory is neuroplasticity. Structural hippocampal plasticity is provided by changes in spine formation and by adult neurogenesis in the subgranular zone (SGZ) of the DG [127,128,129] which is strongly influenced by the circadian system [130]. We have shown that adult neurogenesis is affected in BMAL1-deficient mice [59,131,132]. Moreover, the fine astrocytic processes ensheathing the hippocampal mossy fibre synapse and the astrocyte actin cytoskeleton are affected in BMAL1-deficient mice [133], indicating that the molecular clockwork modulates astrocyte-neuron interaction at the structural level of the tripartite synapse. Consistently, both neurons and astrocytes show time-of-day-dependent structural and functional changes in CA1, while pyramidal neurons change the surface expression of NMDA receptors, and astrocytes change the proximity to synapses [80]. Interestingly, the activation of puringergic receptors by extracellular ATP plays an important role in neuron–glia interaction and modulates synaptic strength (reviewed in [134]). Various purinergic receptors show a time-of-day-dependent oscillation in the hippocampus [131], some in phase with the SCN [132], suggesting a general regulatory mechanism across brain regions. 

Functional hippocampal neuronal plasticity is provided by long-term potentiation (LTP), defined as a persistent strengthening of glutamatergic synapses based on recent patterns of activity [135]. Mice with mutations of the clock gene *Per2* show changes in the LTP of the Schaffer collateral-CA1 synapse, presumably as a result of the reduced activation of the CREB [48], which is implicated in LTP and memory formation (reviewed in [136]). Signal transduction pathways, including MAPK and CREB are implicated in amygdala- and hippocampus-dependent long-term memory consolidation in the context of fear conditioning [137,138]. In mice, MAPK activation shows a circadian oscillation with high levels during the (subjective) light phase, associated with an increased consolidation of contextual fear memory during the (subjective) light phase [139]. Similarly, mice show time-of-day-dependent changes in hippocampus-dependent memory formation [119] and retrieval [43]. Consistently, time-of-day-dependent changes in LTP occur in the rodent hippocampus (reviewed in [140]) [80]. Nocturnal rodents exposed to chronic phase shifts show a deficit in spatial learning and memory, indicating that chronodisruption affects hippocampal function [141,142]. In hamsters, this is associated with impaired adult neurogenesis and is independent of systemic glucocorticoids [142]. In Bmal1-deficient mice, impaired contextual fear and spatial memory are associated with the reduced activation of the MAPK signalling pathway [57], indicating an interconnection between the molecular clockwork and pathways implicated in the memory consolidation. Importantly, LTP is decreased in hippocampal slices from BMAL1-deficient mice, indicating that the hippocampal molecular clockwork modulates functional synaptic plasticity. In mice with a hippocampus-specific inhibition of BMAL1 function (dnBMAL1), showing a normal circadian rhythm of locomotor activity, memory retrieval is impaired [43]. This indicates that the molecular clockwork contributes to hippocampus-dependent learning. Remarkably, the retrieval deficits observed in dnBMAL1 mice seem to be due to impaired dopamine D1 and D5 receptor-dependent cAMP signal transduction [43]. In addition, phosphorylation of the AMPA-type glutamate receptor subunit GluA1, which is modulated by D1/D5 dopamine receptor activation [143], regulates AMPA receptor trafficking, and is suggested to play a crucial role in hippocampus-dependent learning and memory [144], which is reduced in dnBMAL1 mice. Hence, the molecular clockwork might control rhythms in hippocampus-dependent memory function via cAMP-dependent D1/D5 dopamine receptor signal transduction and GluA1 phosphorylation. Importantly, dopamine, which plays important roles in executive functions, motor control, motivation, arousal, reinforcement, and reward, seems to be an important mediator in maintaining circadian rhythms in many brain regions, and the loss of dopamine neurons might account for the impairment of circadian rhythms in Parkinson’s disease (reviewed in [145,146,147]). In the SCN, D1 receptor signalling is necessary for photoentrainment [148], a mechanism that employs similar signal transduction pathways as hippocampus-dependent memory consolidation. In addition, other neurotransmitters as well as hormones might have time-of-day-dependent and/or circadian effects on cognitive performance and memory formation. The rhythm in locomotor activity levels does not show a clear correlation with circadian rhythms in memory formation in various species (reviewed in [140]). Microdialysis experiments show a time-of-day-dependent fluctuation in basal levels of various neurotransmitters in the hippocampus of nocturnal rodents. Basal levels of adenosine, noradrenalin, acetylcholine, and serotonin are higher during the dark phase compared to the light phase [149]. These neurotransmitters are key regulators of synaptic plasticity in the hippocampus [150,151,152,153]. Acetylcholine release shows a strong correlation with locomotor activity [154] and the activity of thyroid hormones, which is also implicated in hippocampus-dependent learning (reviewed in [155]). Furthermore, in a recent study by McCauley and co-workers, corticosterone was identified as a key factor in regulating time-of-day-dependent changes in synaptic strength [80]. 

A brief light pulse applied during the dark phase enhances the consolidation of contextual fear conditioning and CA1 LTP [156], indicating that light has a strong impact on contextual fear learning in nocturnal animals. Although there is evidence that (blue) light also enhances alertness and cognitive function in humans (reviewed in [157]), the mechanism might be different from nocturnal rodents, where light represents a strong aversive stimulus. So far, little is known on the neuronal network transmitting non-visual photic information to the hippocampus. Anterograde polysynaptic tracing of retino-recipient regions identified the amygdala and the hippocampal CA1 region among many others [158]. Data by Richetto et al. [121] suggest that mesolimbic structures, such as the nucleus accumbens and the midbrain might be involved in the effect of the light phase on behavioural responses [121]. Interestingly, the chemogenetic activation of ipRGCs in dark-adapted mice evokes circadian phase resetting and increases anxiety-related behaviour similar to light exposure [159]. Moreover, it induces neuronal activation in various brain regions, including the amygdala and the unspecific thalamus, which are implicated in anxiety and arousal, respectively [159]. Thus, non-visual light information affects alertness and anxiety presumably via the unspecific thalamus and the amygdala. Interestingly, distinct ipRGC projections mediate the effects of light on learning and mood (Figure 3). The projections of the ipRGCs to the SCN mediate the effects of light on learning, which are independent of the SCN function in circadian rhythm generation. The nature of this pathway is unknown so far but may include projections of the SCN to other hypothalamic nuclei and the septal region [160] known to project to the hippocampus [9,161]. SCN-independent projections to the thalamic perihabenular nucleus drive the effects of light on emotional behaviour [11]. The perihabenular nucleus projects to the ventromedial prefrontal cortex, which is implicated in the processing of risk and fear upstream of the amygdala and in the consolidation of extinction learning [162], to the dorsomedial striatum, which is implicated in motor learning and performance (reviewed in [163]), and to the nucleus accumbens, which integrates input from the prefrontal cortex, amygdala, ventral hippocampus, and from the dopaminergic neurons of the ventral tegmental nucleus and plays a significant role in the processing of motivation, aversion, reward, and reinforcement learning and in the induction of slow-wave sleep [164]. Hence, the perihabenular nucleus seems to play an important role in mediating the effects of light on emotion/mood implicated in cognitive function and learning. This might also be relevant for the effects of light on cognition and mood in humans. In patients with major depressive disorder and bipolar depression, treatment with bright white light (>5000 lux, >30 min) during the day, known as light therapy, ameliorates the symptoms [165]. On the other hand, exposure to excessive light at night shortly before bedtime, is associated with a greater risk for depressive symptoms (reviewed in [166]). In mice, light at night induces depressive-like behaviour without disturbing circadian rhythms [167]. This effect was mediated by a projection from the ipRGCs to the perihabenular nucleus and from here to the nucleus accumbens, suggesting that this brain circuitry might also be relevant for mental health effects of the prevalent night-time illumination in the modern 24/7 society [167]. In addition, the superior colliculus, which receives direct retinal input and mediates behaviour responses to visual danger signals, projects to the reticular formation [168] and to the amygdala via the thalamic pulvinar, driving emotional responses to visual information [169]. However, little is known about the relevance of these projections for light-at-night-induced changes in brain plasticity. 

### 2.6. Neuropathological Conditions and Circadian Misalignment

Chronic disruption of circadian rhythms in humans, for example in shift and night work, has aversive effects on health in general (reviewed in [170]) and may even have an effect on preterm birth (reviewed in [171]). Especially excessive artificial light at night, e.g., in shift work, during repeated transmeridian travel, or from the use of illuminated electronic devices, such as mobile phones, televisions, and personal computers disrupts circadian rhythms and suppresses the production of melatonin, both these changes are light intensity and wavelength dependent [172]. In recent years, significant technical advances have been made in developing blue-free white light-emitting diodes [173], or blue light filters, such as in the night-shift mode of smartphones, that can help to prevent chronodisruption and to preserve rhythmic melatonin production. Chronodisruption is associated with higher risk for brain dysfunction, such as sleep disturbances, impaired alertness, and depression (reviewed in [174]), and impairs brain plasticity. Moreover, there is a reciprocal relationship of chronodisruption and neurological or psychiatric conditions and diseases. Psychiatric conditions, such as depressive disorders, bipolar disorder, seasonal affective disorder, and schizophrenia, are frequently associated with abnormalities in the sleep/wake cycle or in social rhythms [175,176]. Patients with Alzheimer’s disease (AD) suffer from circadian disruption, while the cause-and-effect relationship is still unclear (reviewed in [177,178]). Moreover, in mouse models for AD [179] and other synucleopathies and neurodegenerative diseases [180,181], the light input into the circadian system and thus masking and/or entrainment is impaired, thus further enhancing circadian misalignment. Hence, there is an interconnection between neuropathological conditions and the circadian system at various levels.

## 3. Summary

Rhythmic brain function is controlled by light and the circadian system. The SCN and its rhythmic output govern endogenous/circadian rhythms, which are entrained to the environmental light/dark cycle by retinal input. Direct and indirect retinal input is provided to many parts of the brain, contributing to light-dependent responses in a wide range of neuronal networks. At the cellular level, synaptic plasticity is modulated by a molecular clockwork in neurons and glia, which are modulated by various rhythmic neuronal, glial, and endocrine signals.

## 4. Conclusions and Outlook

In recent decades, great advances have been made in understanding the molecular basis of circadian time-keeping mechanisms and circadian light perception in the mammalian brain. The greatest challenge for the future will be to decipher the complex interactions and connectivity between the various components of the circadian system at the level of complex neural networks. This is mandatory for understanding not only rhythmic basic brain function, such as sleep but also higher cognitive function such as learning and memory under physiological and pathological conditions. 

The cholinergic ascending reticular activating system (ARAS) is a key element in the control of wakefulness and sleep–wake transition. It activates the thalamus, which controls general cortical activity and transmission of sensory information to the cerebral cortex. By interacting with other brain stem reticular nuclei, the ARAS also modulates muscle tone as well as autonomic functions during wake and sleep. In addition, cortical activity is indirectly (via the ARAS) and directly (not shown) modulated by the noradrenergic locus coeruleus, the dopaminergic ventral tegmental area, and the serotoninergic raphe nuclei. Sleep- and wake-inducing hypothalamic nuclei control ARAS activity. During sleep and wake, the ARAS is inhibited and activated by a system of GABAergic neurons in the ventrolateral preoptic nucleus (VLPO) and of histaminergic neurons in the tuberomammillary nucleus (TMN), respectively. Orexinergic neurons in the lateral hypothalamus (LH) contribute to arousal projections into the TMN, the forebrain, the thalamus, and the brain stem. The circadian rhythm in sleep/wakefulness is controlled by the suprachiasmatic nucleus (SCN) which projects to the dorsomedial hypothalamus (DMH) via the subparaventricular zone (SPZ). DMH neurons project to the VLPO and the lateral hypothalamus using inhibitory and excitatory neurotransmitters orchestrating rhythmic changes in sleep–wake and wake–sleep transitions. Light information reaches the SCN, the SPZ, the LH, and the VLPO by direct innervation from the retina. Diencephalic, mesencephalic and telencephalic brain regions are assembled in red, grey, and green boxes, respectively. ACh, acetylcholine. Based on [103,104,105,113].

## Figures and Tables

**Figure 1 ijms-23-02778-f001:**
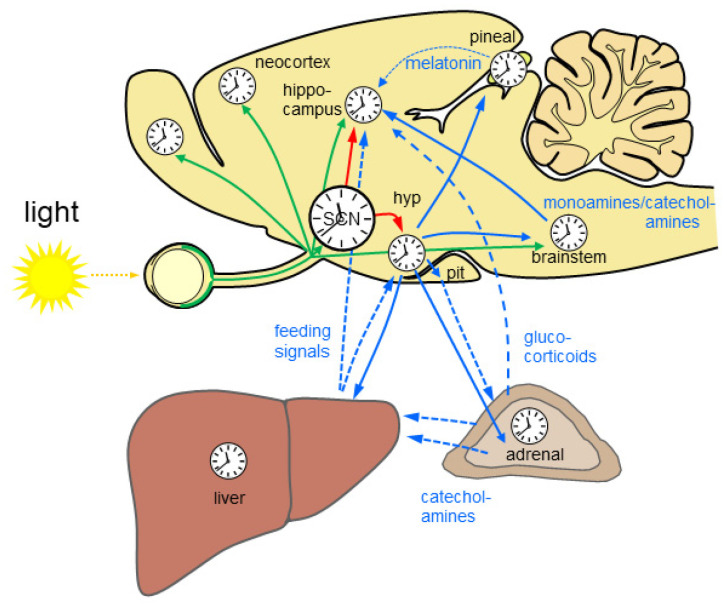
The mammalian circadian system is highly complex and hierarchically organized. Almost all brain regions and organs comprise a molecular clockwork (clocks) which controls rhythmic cell function. Rhythmic light information is provided directly and indirectly to many brain regions (green arrows) and drives time-of-day-dependent rhythms in brain and periphery. The central circadian rhythm generator which is located in the suprachiasmatic nucleus (SCN) of the hypothalamus is entrained by light. SCN lesion results in loss of circadian rhythms. Rhythmic output of the SCN governs subsidiary circadian oscillators in the brain (red arrows). Different nuclei in the hypothalamus (hyp) control rhythmic physiology and behavior via neuronal connections including the autonomous nervous system (blue solid arrows) and endocrine signals (blue dashed arrows) via the pituitary (pit). Rhythmic endocrine signal from the pineal gland and the periphery (blue dashed lines) provide additional rhythmic signals for the brain. The liver is depicted exemplarily for the gastrointestinal system. Monoamines and catecholamines from the brain stem provide important rhythmic drive for alertness and motivation at the level of the forebrain. Based on [9,10,11].

**Figure 2 ijms-23-02778-f002:**
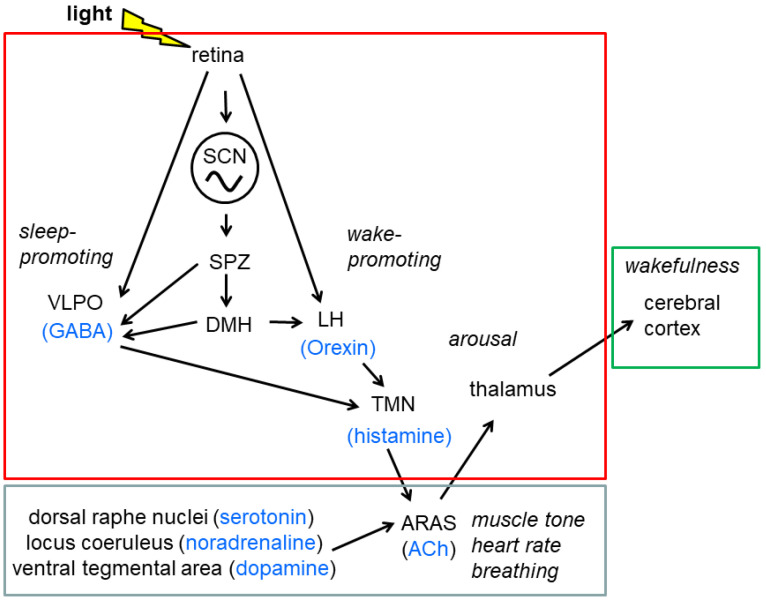
Simplified summary of the effects of light and the suprachiasmatic nucleus (SCN) on the brain circuitry that governs sleep and wakefulness. The cholinergic ascending reticular activating system (ARAS) is a key element in the control of wakefulness and sleep–wake transition. It activates the thalamus, which controls general cortical activity and transmission of sensory information to the cerebral cortex. By interacting with other brain stem reticular nuclei, the ARAS also modulates muscle tone as well as autonomic functions during wake and sleep. In addition, cortical activity is indirectly (via the ARAS) and directly (not shown) modulated by a variety of brain stem nuclei, which employ different neurotransmitters, including the noradrenergic locus coeruleus, the dopaminergic ventral tegmental area, and the serotoninergic raphe nuclei. Sleep- and wake-inducing hypothalamic nuclei control ARAS activity. During sleep and wake, the ARAS is inhibited and activated by a system of GABAergic neurons in the ventrolateral preoptic nucleus (VLPO) and of histaminergic neurons in the tuberomamillary nucleus (TMN), respectively. Orexinergic neurons in the lateral hypothalamus (LH) contribute to arousal by projections into the TMN, the forebrain, the thalamus, and the brain stem. The circadian rhythm in sleep/wakefulness is controlled by the suprachiasmatic nucleus (SCN) which projects to the dorsomedial hypothalamus (DMH) via the subparaventricular zone (SPZ). DMH neurons project to the VLPO and the lateral hypothalamus using inhibitory and excitatory neurotransmitters orchestrating rhythmic changes in sleep–wake and wake–sleep transitions. Importantly, the SCN, the SPZ, the LH, and the VLPO receive direct innervation from the retina. Diencephalic, brain stem, and telencephalic brain regions are assembled in red, grey and green boxes, respectively. ACh, acetylcholine. Based on [103,104,105,113].

**Figure 3 ijms-23-02778-f003:**
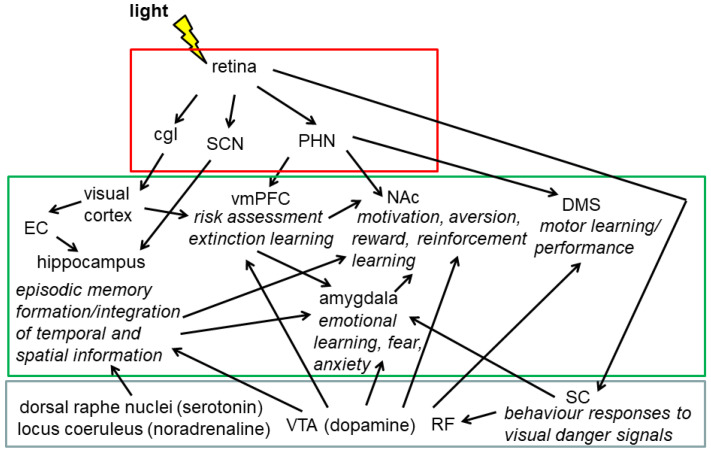
Simplified summary of the effects of light and the suprachiasmatic nucleus (SCN) on the major brain circuitry responsible for emotion and learning.Projections of the retina to the SCN mediate the effects of light on learning presumably via indirect projections to the hippocampus. Projections of the retina to the perihabenular nucleus (PHN) mediates effects of light on emotion/mood, memory consolidation, and motor learning. The PHN projects to the ventromedial prefrontal cortex (vmPFC) and the nucleus accumbens (NAc), both are closely interconnected with the amygdala. The NAc integrates input from the vmPFC, amygdala, hippocampus and from dopaminergic neurons of the ventral tegmental nucleus (VTA). The VTA and other monoaminergic nuclei of the reticular formation (RF) project to various brain regions, including those related to learning and memory, providing emotional and motivational drive. The superior colliculus (SC) receives direct retinal input and projects to the RF and to the amygdala via the thalamic pulvinar (not shown). Visual information is transmitted from the retina to the visual cortex via the corpus geniculatum laterale (cgl) and from there to most of the cerebral cortex including the hippocampus via the entorhinal cortex (EC). Diencephalic, brain stem, and telencephalic brain regions are assembled in red, grey, and green boxes, respectively. DMS, dorsomedial striatum.

## Data Availability

Not applicable.

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
