# Peer review of "The Effects of Light and the Circadian System on Rhythmic Brain Function"

_ijms, 2022, doi:10.3390/ijms23052778_

Round 1

Reviewer 1 Report

Present MS of Professor Gall is focused on effects of light and the circadian system on rhythmic brain functions. The topic is interesting and up to date and the author provide long list of references. Unfortunately, I am very sorry to say, that MS in not very informative and does not provide useful synthesis of information. It is really rather a list of papers that should be read to understand the topic that anything else. It is difficult to read MS as there are too many links to external sources and a lot of vague sentences; e.g.

Glucocorticoids have been shown to control various subsidiary clocks in the periphery, such as the liver, kidney, and heart …

What does the reader really learn from this sentence?

Major comments:

As the synthesis of information is scarce this article is not helpful in understanding the topic.

The main aim of review – to perform synthesis and help reader to understand the topic – was not fulfilled.

Out of 179 references 46 are links to review of some else. It is too much. If someone would really act according advice of author he or she would spend several weeks to acquire the topic. This not a purpose of review. A good review is comprehensive and readable text that carry reader through the complicated subject in such a way, that it more convenient than reading 50 other articles (like in this case).

Moreover, there are only two pictures. Unfortunately those pictures are not supported by evidences. Especially figure 2 misses referenced to literature that would support suggested links.

I strongly suggest add at least 3-4 more figures and/or tables that would facilitate understanding of topic and provide synthesis. Especially hypothalamic nuclei deserve picture. On the other hand, knockouts models deserve table with all known evidences, not just some for example.

Information from figures 1 and 2 must be supported by particular references.

Knowledge about light perception in the brain is not that new, all references should be included, especially when it is in the title of the article.

Minor comment:

What does mean 8 in the title?

Author Response

Reviewer 1

Present MS of Professor Gall is focused on effects of light and the circadian system on rhythmic brain functions. The topic is interesting and up to date and the author provide long list of references. Unfortunately, I am very sorry to say, that MS in not very informative and does not provide useful synthesis of information. It is really rather a list of papers that should be read to understand the topic that anything else. It is difficult to read MS as there are too many links to external sources and a lot of vague sentences; e.g.

Glucocorticoids have been shown to control various subsidiary clocks in the periphery, such as the liver, kidney, and heart …

What does the reader really learn from this sentence?

Response: Thank you very much for appreciating that the topic is interesting and up to date.

I’ve made the aim of the review clearer (see also below)

In chapter 2.5.1 the readers learn that rhythmic hormone (e.g. glucocorticoid) secretion is controlled by the circadian clock and affected by light during night. Glucocorticoids in turn control rhythmic function in subsidiary oscillators in periphery and presumably also in brain.

Major comments:

As the synthesis of information is scarce this article is not helpful in understanding the topic.

The main aim of review – to perform synthesis and help reader to understand the topic – was not fulfilled.

Response: I’ve now included the aim of this review in the introduction as follows: The aim of this broad review article is to highlight the role of light on rhythms in physiology and behavior, especially in nocturnal rodents from a neuroanatomical point of view, and to emphasize the important distinction between light-driven/time-of-day-dependent and endogenously-driven/circadian rhythms.

Out of 179 references 46 are links to review of some else. It is too much. If someone would really act according advice of author he or she would spend several weeks to acquire the topic. This not a purpose of review. A good review is comprehensive and readable text that carry reader through the complicated subject in such a way, that it more convenient than reading 50 other articles (like in this case).

Response: There are different concepts of reviews. The one you describe is suitable to summarize one complicated subject. A different concept, I have followed with my review is a broader approach to provide a reference for a complex interplay of different complicated subjects, similar to a compendium. The cited review articles are “further reading” on the different subjects. However, the reader does not have to read all cited references to understand the importance of light and circadian system for rhythmic brain function. I’ve made that concept clear now by stating: “The aim of this broad review article…”.  

Moreover, there are only two pictures. Unfortunately those pictures are not supported by evidences. Especially figure 2 misses referenced to literature that would support suggested links.

I strongly suggest add at least 3-4 more figures and/or tables that would facilitate understanding of topic and provide synthesis. Especially hypothalamic nuclei deserve picture. On the other hand, knockouts models deserve table with all known evidences, not just some for example.

 Information from figures 1 and 2 must be supported by particular references.

Responses:
I’ve now included particular references in figure 1.
I agree that figure 2 might be too complex for readers not well versed in neuroanatomy. Thus, I extracted to two major circuits and provided the respective references.
A list of all known evidence from knockouts models goes beyond the scope of this review
(see aim of this review).

Knowledge about light perception in the brain is not that new, all references should be included, especially when it is in the title of the article.

Response:
I totally agree that knowledge about light perception in the brain is not new. However, the aim of this review is to summarize the current knowledge on light perception in the brain as it is still underappreciated in many studies e.g. on emotion and cognition. I’ve included all references I regarded relevant. Please specify which ones I have omitted if applicable.

Minor comment:

What does mean 8 in the title?

Responses:
I don’t see and 8 in the title.

Reviewer 2 Report

It is a complete review, very well explained, with updated references on a topic of maximum relevance and interest. Nowadays, especially after the pandemic, in which we spend many hours under the influence of artificial light, it is considered crucial to address aspects such as chronodisruption and its effects on human health. In any case, the author should pay attention to the following topic:

  • The author speaks of the passage of humans from their nocturnal character to their diurnal character. It should explain this step in more detail. It would be interesting to dedicate a specific section to it, which would capture the attention and interest of many readers. These questions should be considered: How many years did it take for humans to go from their nocturnal nature to their diurnal nature? How were biological clocks and circadian rhythms reset? What was the competitive advantage of going diurnal? Are human really the only diurnal mammals? … These and other questions should be accompanied by several bibliographical references.
  • Artificial light at night (ALAN) is a serious problem. It would be interesting and enriching to mention the technological efforts to develop free-blue LED luminaires. You could include this recent article [1] published in an MDPI journal.
  1. Menéndez-Velázquez, A.; Morales, D.; García-Delgado, A.B. Light Pollution and Circadian Misalignment: A Healthy, Blue-Free, White Light-Emitting Diode to Avoid Chronodisruption. International Journal of Environmental Research and Public Health 2022, 19, 1849, doi:10.3390/ijerph19031849.

Author Response

It is a complete review, very well explained, with updated references on a topic of maximum relevance and interest. Nowadays, especially after the pandemic, in which we spend many hours under the influence of artificial light, it is considered crucial to address aspects such as chronodisruption and its effects on human health.

Response: Thank you very much for appreciating the importance of the review. 

In any case, the author should pay attention to the following topic:

  • The author speaks of the passage of humans from their nocturnal character to their diurnal character. It should explain this step in more detail. It would be interesting to dedicate a specific section to it, which would capture the attention and interest of many readers. These questions should be considered: How many years did it take for humans to go from their nocturnal nature to their diurnal nature? How were biological clocks and circadian rhythms reset? What was the competitive advantage of going diurnal? Are human really the only diurnal mammals? … These and other questions should be accompanied by several bibliographical references.

Response: In the introduction, I have included some additional information on crepuscular species, diurnal rodents and the flexible of the primate visual system. I’m not an expert on phylogeny, thus it is hard for me to answer all your questions but I’ve included some additional references covering these issues. 

  • Artificial light at night (ALAN) is a serious problem. It would be interesting and enriching to mention the technological efforts to develop free-blue LED luminaires. You could include this recent article [1] published in an MDPI journal.

Response: You are totally right, this is very important.  I’ve included this information in the section on Neuropathological conditions and circadian misalignment (page 12).